



# Global seasonal distribution of CH₂Br₂ and CHBr₃ in the upper troposphere and lower stratosphere

Markus Jesswein[1], Rafael P. Fernandez[2], Lucas Berná[3], Alfonso Saiz-Lopez[4], Jens-Uwe Grooß[5], Ryan Hossaini[6], Eric C. Apel[7], Rebecca S. Hornbrook[7], Elliot L. Atlas[8], Donald R. Blake[9], Stephen Montzka[10], Timo Keber[1], Tanja Schuck[1], Thomas Wagenhäuser[1], and Andreas Engel[1]

[1]University of Frankfurt, Institute for Atmospheric and Environmental Sciences, Frankfurt, Germany
[2]Institute for Interdisciplinary Science (ICB), National Research Council (CONICET), FCEN-UNCuyo, Mendoza, Argentina
[3]Atmospheric and Environmental Studies Group (GEAA), National Technological University (UTN-FR Mendoza), Mendoza, Argentina
[4]Department of Atmospheric Chemistry and Climate, Institute of Physical Chemistry Rocasolano, CSIC, Madrid, Spain
[5]Institute of Energy and Climate Research – Stratosphere (IEK-7), Forschungszentrum Jülich, Jülich, Germany
[6]Lancaster Environment Centre, Lancaster University, Lancaster, UK
[7]Atmospheric Chemistry Observations & Modeling Laboratory, National Center for Atmospheric Research, Boulder, CO, USA
[8]University of Miami, Miami, FL, USA
[9]University of California, Irvine, Department of Chemistry, Irvine, CA, USA
[10]Global Monitoring Laboratory, NOAA, Boulder, CO, USA

**Correspondence:** Markus Jesswein (jesswein@iau.uni-frankfurt.de)

**Abstract.**

Bromine released from the decomposition of short-lived brominated source gases contributes as a sink of ozone in the lower stratosphere. The two major contributors are $CH_2Br_2$ and $CHBr_3$. In this study, we investigate the global seasonal distribution of these two substances, based on four High Altitude and Long Range Research Aircraft (HALO) missions, the HIAPER

5 Pole-to-Pole Observations (HIPPO) mission, and the Atmospheric Tomography (ATom) mission. Observations of $CH_2Br_2$ in the free and upper troposphere indicate a pronounced seasonality in both hemispheres, with slightly larger mixing ratios in the Northern Hemisphere (NH). Compared to $CH_2Br_2$, $CHBr_3$ in these regions shows larger variability and less clear seasonality, presenting larger mixing ratios in winter and autumn in NH mid to high latitudes. A clear $CH_2Br_2$ maximum is observed in the NH during autumn with a less pronounced similar feature in the Southern Hemisphere (SH). This suggests that transport

10 processes may be different in both hemispheric autumn seasons, which implies that the influx of tropospheric air ("flushing") into the NH lowermost stratosphere is more efficient than in the SH. However, the SH database is insufficient to quantify this difference. We further compare the observations to model estimates of TOMCAT and CAM-Chem, both using the same emission inventory. The pronounced tropospheric seasonality of $CH_2Br_2$ in the SH is not reproduced by the models, presumably due to erroneous seasonal emissions or atmospheric photochemical decomposition efficiencies. In contrast, model simulations

15 of $CHBr_3$ show a pronounced seasonality in both hemispheres, which are not confirmed by observations. The distributions of both species in the lowermost stratosphere of the Northern and Southern Hemispheres are overall well captured by the models with the exception of southern hemispheric autumn, where both models present a bias that maximizes in the lowest 40 K above





the tropopause, with considerably lower mixing ratios in the observations. Thus, both models reproduce equivalent "flushing" in both hemispheres, which is not confirmed by the available observations. Our study emphasises the need for more extensive observations in the SH to fully understand the impact of $CH_2Br_2$ and $CHBr_3$ on lowermost stratospheric ozone loss and to help constraining emissions.

## 1   Introduction

Reactive gases containing chlorine and bromine are very efficient in destroying stratospheric ozone in catalytic reaction cycles. The relative efficiency of bromine is 60–65 times higher than that of chlorine (e.g., Sinnhuber et al., 2009; WMO, 2018). A more recent study reports a 74 times higher efficiency of bromine (Klobas et al., 2020). Thus, although the amount of bromine in the stratosphere is much smaller than that of chlorine, bromine plays an important role in stratospheric ozone chemistry. Major contributors to the stratospheric bromine are the four major halons H-1211 ($CBrClF_2$), H-1301 ($CBrF_3$), H-1202 ($CBr_2F_2$), and H-2402 ($CBrF_2CBrF_2$), all originating from anthropogenic sources. Furthermore, methyl bromide ($CH_3Br$) is a major contributor, which has both natural and anthropogenic sources. Additionally, the so-called "very-short-lived substances" (VSLS), with lifetimes shorter than 6 month, can contribute bromine to the stratosphere and thus lead to ozone loss. Bromine VSLS (Br-VSLS in the following) contributed to about a quarter to stratospheric bromine in 2016 with a total of 5 (3–7) ppt (parts per trillion)  (Engel and Rigby, 2018). The contribution is partly in form of organic source gases (source gases injection; SGI) providing 2.2 (0.8–4.2) ppt Br and in inorganic form as photochemical decomposed species (product gases injection; PGI) with 2.7 (1.7–4.2) ppt Br  (Engel and Rigby, 2018). Once in the lowermost stratosphere (LMS), released bromine from VSLS can affect the ozone abundance and distribution. Especially in the mid latitude LMS, bounded by the 380 K potential temperature surface at the top and the extratropical tropopause at the bottom (e.g., Hoor et al., 2005), bromine-driven ozone loss cycles gain importance (e.g., Daniel et al., 1999; Salawitch et al., 2005). It is also a region where ozone changes have a relatively large radiative effect (Hossaini et al., 2015, and references therein).

Transport of source gases into the LMS can occur via different pathways. Transport associated with the global scale stratospheric Brewer-Dobson circulation (BDC) brings older air from the stratospheric overworld into the upper troposphere and lower stratosphere (UTLS) via the deep branch with long transit times, as well as air from the tropics and subtropics into the UTLS via the shallow branch with shorter transit times (Birner and Bönisch, 2011). On the other hand, air can be transported directly into the LMS via the extratropical tropopause by troposphere-to-stratosphere transport (TST). Kunkel et al. (2019) describes in more detail the processes for stratosphere-troposphere exchange (STE) in the mid latitudes, for example due to Rossby wave breaking and tropopause folds along jet streams. Furthermore, Kunkel et al. (2019) suggested that air masses potentially enter the stratosphere in ridges of baroclinic waves at the anticyclonic side of the jets above the outflow of warm conveyor belts (whereby the significance of this process still needs to be assessed). Previous studies estimated that the contribution of extratropical tropospheric air to the LMS shows a pronounced seasonality. Hoor et al. (2005) used CO in situ measurements to infer a fraction of 35% extratropical air in winter and spring LMS composition over Europe, whereas the fraction rises to 55% in summer and autumn. A similar seasonality but with much higher extratropical fractions was found





by Bönisch et al. (2009) using in situ measurements of $CO_2$ and $SF_6$. Extratropical tropospheric fractions up to 90% were found in October and lowest fraction below 20% in April. Hegglin et al. (2009) used $O_3$, $H_2O$, and CO measurements from the Atmospheric Chemistry Experiment Fourier Transformation Spectrometer (ACE-FTS) on Canada's SCISAT-1 satellite to investigate the global behavior of the extratropical tropopause transition layer (ExTL), which is the finite chemical transition layer across the tropopause and into the LMS. Major findings were a shallower transition layer in the Southern Hemisphere (SH) with a weaker troposphere-stratosphere-transport compared to the Northern Hemisphere (NH) and an overall smaller seasonal variation. Hegglin and Shepherd (2007) showed that "flushing" of the LMS with younger air from the tropics is most evident in NH summer and autumn and is weaker in the SH. These results are confirmed by the CO tracer distribution in Hegglin et al. (2009).

The different transport paths have an influence on the distribution of Br-VSLS, especially in the LMS of each hemisphere. This study focuses on bromoform ($CHBr_3$) and dibromomenthane ($CH_2Br_2$), which are the most abundant Br-VSLS. The local lifetime of $CH_2Br_2$ ranges from 150 to 890 days and for $CHBr_3$ between 17 to 88 days, depending on location and season (see Sect. 4, Table 3 for seasonally resolved local lifetimes). Main sources of these Br-VSLS are open ocean and coastal regions via the metabolism of marine organisms such as phytoplankton and macro-algae (e.g., Carpenter and Liss, 2000; Quack et al., 2007; Leedham et al., 2013). Sturges et al. (1993) and Abrahamsson et al. (2018) suggested that winter sea ice could potentially be an additional source of Br-VSLS. Athropogenic sources are water chlorination (e.g., Worton et al., 2006; Maas et al., 2021) and industrial discharge of chlorinated effluents to seawater (Quivet et al., 2022; Maas et al., 2019; Hamed et al., 2017; Boudjellaba et al., 2016; Sam Yang, 2001). The contribution of treated water may have an impact on local scale only, and the significance of theses sources on a global scale remains unclear (e.g., Quivet et al., 2022; Liu et al., 2011). Studies of the observation-based distribution, especially when looking at stratospheric input of Br-VSLS focused predominantly on the tropics and the NH. The current best estimates of tropical tropopause values of $CH_2Br_2$ and $CHBr_3$ are given in Engel and Rigby (2018) ranging from 0.81 (0.59–0.98) ppt to 0.64 (0.32–0.89) ppt $CH_2Br_2$ from the level of zero clear-sky radiative heating (LZRH) to the tropical tropopause (TTP), and 0.36 (0.05–0.72) ppt to 0.19 (0.01–0.54) ppt $CHBr_3$ from LZRH to TTP (Wofsy, 2011; Sala et al., 2014; Navarro et al., 2015; Pan et al., 2017). A recent study by Keber et al. (2020) reported aircraft measurements of Br-VSLS at the tropopause and LMS in NH mid to high latitudes during winter and late summer to early autumn. They reported systematically higher mixing ratios of $CHBr_3$ at the extratropical tropopause than those at the TTP. A similar, although less pronounced feature was found for $CH_2Br_2$. This increase was more pronounced in winter, when lifetimes increase at higher latitudes. In addition, Keber et al. (2020) compared their observations with model estimates using different emission scenarios. Although no scenario was able to capture the tropical and extratropical values from their observations, the Ordóñez et al. (2012) scenario showed an overall good agreement, especially for $CH_2Br_2$.

As Keber et al. (2020) already pointed out, there are still some knowledge gaps regarding the distribution of the Br-VSLS in the upper atmosphere, as they show only observations of the NH in winter and late summer to early autumn. Especially the data coverage in the SH is sparse. It is expected that the distribution of the Br-VSLS in the SH may differ from the NH distribution, due to fewer source regions like coastal ocean regions. Here we expand the analysis of Keber et al. (2020) to a global view of the two major Br-VSLS. For that, besides using the observations already used in Keber et al. (2020), namely the



High Altitude and Long Range Research Aircraft (HALO) missions TACTS, PGS, and WISE, we also use observations from the southern hemispheric HALO mission SouthTRAC from September to November 2019. Furthermore, we use observations from the HIAPER Pole-to-Pole Observations (HIPPO) mission and the Atmospheric Tomography (ATom) mission, both of which include data from the Northern and Southern Hemisphere, to investigate the global seasonal distribution of the Br-

VSLS. Observations are compared with two global models, namely CAM-Chem and TOMCAT, both using the same emission scenario of Ordóñez et al. (2012). In Sect. 2, we give a brief overview of the missions and instruments used for this analysis, followed by an introduction to the meteorological data and the models against which we compare the observations in Sect. 3. The distribution of $CH_2Br_2$ and $CHBr_3$ from observations and model simulations are discussed in Sect. 4. We start with the broader global distribution by presenting seasonal zonal mean mixing ratios from both hemispheres from ground to the lower

stratosphere, moving on to a closer look at near-tropopause mixing ratios, and finally focus on the vertical distribution in the mid latitudes of NH and SH. Lastly, we summarize the conclusion and provide an outlook in Sect. 5.

## 2 Measurements

### 2.1 HALO missions

In this work, we use data from four missions conducted with the High Altitude and Long Range Research Aircraft (HALO).

HALO is a Gulfstream V (GV) aircraft and can reach altitudes up to 15 km. The first mission is the TACTS (Transport and Composition in the Upper Troposphere/Lowermost Stratosphere) mission, conducted in August and September 2012 with flights covering an area from the Cape Verde islands to the Norwegian archipelago of Spitsbergen and over Europe and the Atlantic Ocean. The base of all flights was Oberpfaffenhofen (Germany) (Fig. 1, blue tracks). The second mission was PGS, consisting of three sub-missions: POLSTRACC (Polar Stratosphere in a Changing Climate), GW-LCYCLE (Investigation of

the Life cycle of gravity waves) and SALSA (Seasonality of Air mass transport and origin in the Lowermost Stratosphere). The mission took place from December 2015 to March 2016 with flights mainly in the Arctic and covering Greenland, the North Atlantic, and Europe. Flights were conducted from Oberpfaffenhofen (Germany) and from Kiruna (Sweden) (Fig. 1, purple tracks) (Oelhaf et al., 2019). The third mission was the WISE (Wave-driven ISentropic Exchange) mission between September and October 2017. Flights were conducted mainly from Shannon (Ireland), covering an area above the Atlantic Ocean and

Western Europe (Fig. 1, green tracks). Finally, the SouthTRAC (Southern Hemisphere Transport, Dynamics, and Chemistry) mission took place from September to November 2019. It is the only one of the four HALO missions that covers the SH. In addition to the scientific transfer flights, which departed from Oberpfaffenhofen (Germany) via the Cape Verde islands to South America, all other flights took place from Rio Grande (Argentina). Flights of the SouthTRAC mission cover the southern Pacific and southern Atlantic oceans near South America and Antarctica (Fig. 1, orange tracks).

Data from the Gas chromatograph for Observational Studies using Tracers (GhOST) in situ instrument were used in this analysis. The instrument has two channels. The first channel couples an isothermally operated gas chromatograph (GC) with an electron capture detection (ECD) (GhOST-ECD) and the second channel couples a temperature programmed GC with a quadrupole mass spectrometer (MS) (GhOST-MS) (see Jesswein et al. (2021) and references therein). For the SouthTRAC





campaign, the GhOST-MS ionization mode was changed from negative chemical ionization (NCI) to electron impact ionization
(EI) to record a broader mass spectra, leading to different detection limits for the Br-VSLS compared to previous campaigns.
In this work, only measurements of the MS channels are used. The measurements of $CH_2Br_2$ and $CHBr_3$ are on NOAA–2003
scale, thus consistent with NOAA/ESRL observations.

## 2.2 HIPPO mission

The HIPPO mission measured cross sections of trace gases over the Pacific Basin and North American mainland (170° W–
80° W) (Fig. 1 black tracks), covering a latitudinal range from the North Pole (85°N) to the coastal region of Antarctica (65° S)
(Wofsy, 2011). The mission was split into seasonal segmented deployments, which took place in January 2009 (HIPPO-
1), October to November 2009 (HIPPO-2), March to April 2010 (HIPPO-3), June to July 2011 (HIPPO-4), and August to
September 2011 (HIPPO-5). The platform used for the observations was the NSF/NCAR High-performance Instrumented
Airborne Platform for Environmental Research (HIAPER) Gulfstream V (GV) aircraft.
Data from two Whole Air Samplers (WAS) were combined in this analysis. The University of Miami operated the Advanced
Whole Air Sampler (AWAS), storing air samples in pressurised stainless steel canisters (Atlas, 2016). The second sampler
operated during HIPPO was the NOAA Whole Air Sampler (NWAS), which stores samples in pressurised glass flasks. Subse-
quently, the samples were analysed using ground-based laboratory GC–MS (gas chromatograph–mass spectrometer) systems.
Results from both laboratories were provided on a scale consistent with NOAA/ESRL ground-based station results (see Hos-
saini et al., 2013, 2016).

## 2.3 ATom mission

The ATom mission was split into four parts, which took place in July to August 2016 (ATom-1), January to February 2017
(ATom-2), September to October 2017 (ATom-3) and April to May 2018 (ATom-4). Thus, all seasons were covered within
a two year time period. In each season, flights started and ended in Palmdale (California, USA) with a route to the western
Arctic, south to the South Pacific, east to the Atlantic, north to Greenland and return across central North America (Wofsy
et al., 2021) (Fig. 1 red tracks). The platform for the ATom mission was the NASA DC-8 aircraft, which is capable of reaching
an altitude of around 12 km.
  As with the HIPPO mission, data from two WAS were used for this analysis. The University of California–Irvine (UC–
Irvine) research group operated one WAS, storing air in stainless steel canisters. Samples were analyzed in the laboratory using
GC with flame ionization detection (FID), ECD, and MS (Barletta et al., 2019). In addition, the NOAA/GML's Programmable
Flask Package Whole Air Sampler (PFP) was operated using glass flasks. The air samples were analysed at the NOAA's Global
Monitoring Division laboratory for trace gases by GC-MS and at the Institute of Arctic and Alpine Research (INSTAAR) Stable
Isotope Lab laboratory for isotopes of methane. Beside the two WAS, the NCAR Trace Organic Gas Analyzer (TOGA) was
operated during ATom. TOGA is an in situ instrument, combining a GC with a MS (Apel et al., 2015), similar to the GhOST-
MS. Asher et al. (2019) used ATom-2 observations among others to improve estimates of short-lived halocarbon emissions
during summer from the Southern Ocean using airborne observations. The two Whole Air Samplers and the TOGA instrument





shared approximately half of the sampling period, generally presenting a good correlation and consistency in mole fractions for $CH_2Br_2$ and $CHBr_3$ (Asher et al., 2019).

## 3 Models and meteorological data

### 3.1 TOMCAT

The TOMCAT (Toulouse Off-line Model of Chemistry And Transport) is an Eulerian offline three-dimensional chemistry transport model (CTM) (Chipperfield, 2006; Monks et al., 2017). The model uses a hybrid vertical sigma-pressure coordinate $(\sigma - p)$ with 60 vertical levels from the ground up to around 60 km. The horizontal resolution was set to $2.8\,° \times 2.8\,°$ (latitude $\times$ longitude). The CTM is forced by meteorological fields (winds, temperature, and humidity) taken from the European Centre for 160 Medium-Range Weather Forecasts ERA5 reanalysis (Hersbach et al., 2020). The internal model step was 30 min and monthly means of the tracers are generated for this study. A similar setup was previously used to study NH Br-VSLS in Keber et al. (2020), beside using ERA-Interim data instead of ERA5. In addition, the configuration used here reads an offline monthly-varying climatological OH concentration field, developed for the TransCom-$CH_4$ project (Patra et al., 2011). In this study, the VSLS emission scenario of Ordóñez et al. (2012) was used with TOMCAT. Model output is available for the period from 2009 165 to 2019.

### 3.2 CAM-Chem

CAM-Chem (Community Atmosphere Model with Chemistry, version 4) is a three-dimensional chemistry climate model (CCM) and a component of the NCAR Community Earth System Model (CESM) (Lamarque et al., 2012). The WACCM physics module for the stratosphere is included and it uses the chemical mechanism of MOZART with different possibilities 170 of complexity for tropospheric and stratospheric chemistry. The model includes a detailed treatment of tropospheric Br-VSLS sources and chemistry described in Fernandez et al. (2014) and Fernandez et al. (2017). The horizontal resolution was set to $0.96\,° \times 1.25\,°$ (latitude $\times$ longitude) and 56 hybrid vertical levels from the surface to around 40 km. The model setup is similar to the one used in Navarro et al. (2015), using the NASA Goddard Global Modeling and Assimilation Office (GMAO) GEOS5 generated meteorology. The model step was 5 min but monthly means of the tracers are used for this study. As with 175 the TOMCAT model, the emission scenario of Ordóñez et al. (2012) was used, with fixed emissions of the VSLS during the whole modelling period (available from 2009–2019).

### 3.3 Meteorological data

Airplanes modified for scientific observations are equipped with on-board instruments to gather meteorological and aircraft parameters along the flight tracks. In addition, local tropopause information along the flight tracks were derived from ECMWF 180 reanalyses. For the SouthTRAC, PGS, HIPPO, and ATom mission, the underlying meteorological field are taken from ERA5 reanalysis, whereas for TACTS and WISE, the underlying fields are from ERA-Interim reanalysis (Dee et al., 2011).





For this work, the potential vorticity (PV)-based dynamical tropopause is used (e.g., Gettelman et al., 2011). The commonly used value of 2 PVU (potential vorticity unit) was used for the dynamical tropopause, this condition was replaced by the potential temperature level of 380 K in the tropics when the 2 PVU level is above (e.g., Keber et al., 2020; Jesswein et al., 2021). We additionally used the same PV-based climatological tropopause information as in Keber et al. (2020), which is based on the ERA-Interim reanalysis.

## 4  Results

In the course of this work, we progressively move from a more global view of the distribution of the tracers to a more detailed view of the UTLS in the mid latitudes of both hemispheres. For the global view, we use latitude as the horizontal coordinate and pressure as the vertical coordinate and thus get a detailed perspective of the troposphere from the ground up to the tropopause and the LMS. As we then look more closely at the area around the tropopause and into the stratosphere, we change the vertical coordinate using potential temperature ($\Theta$) and potential temperature difference to the local tropopause ($\Delta\Theta$) instead of pressure. Transport in the free atmosphere is predominantly isentropic, making $\Theta$ a very useful coordinate. Furthermore, $\Theta$ allows for better vertical resolution as it increases more rapidly with height in stable layers. Finally, the focus moves towards the UTLS of the mid latitudes in the course of the analysis. We switch from the examination in latitude only to a combined coordinate already used in Keber et al. (2020) and Jesswein et al. (2021). Latitude is used for tropospheric observations, whereas equivalent latitude is used for stratospheric ones. The equivalent latitude (Butchart and Remsberg, 1986) is a commonly used horizontal coordinate for studying tracers in the stratosphere and assigns PV to latitude based on the area (of the polar cap) enclosed by the specific isopleth of PV on a given potential temperature contour (Pan et al., 2012). The combined coordinate is referred to as equivalent latitude*.

### 4.1  Altitude-latitude cross sections

We combined the measurements from the different missions, leaving aside that the Br-VSLS may have shown a weak positive trend (e.g., tropical mean $0.017 \pm 0.012$ ppt Br per decade for 1979-2013 from Tegtmeier et al. (2020)). Observations and model results were split by season (DJF: December, January, February; MAM: March, April, May; JJA: June, July, August; SON: September, October, November). Model results are only used for the years and months when observations are available. The data are binned in 10° latitude intervals from 90° S to 90° N. In the vertical we have binned the data between 1000 hPa and 50 hPa into 20 bins. The bin size decreases logarithmically with increasing altitude and thus lower pressure. Thus, the size of the bins varies from about 180 hPa near the ground to 8.5 hPa in the lower stratosphere, which corresponds to an altitude resolution of about 1.3 km.

Figures 2 and 3 show the distributions of $CH_2Br_2$ and $CHBr_3$ for the observations (a–d), TOMCAT model (e–h), CAM-Chem model (i–l), and the differences between the respective model and the observations (m–t). The merged observational dataset allows for a comprehensive representation of the tropospheric distribution except for the southern high latitudes (greater than



70° S) in summer and winter. Furthermore, the LMS of both hemispheres is much better covered by observations in spring and autumn.

The distributions of $CH_2Br_2$ in the troposphere (Fig. 2 a–d) show a general increase in mixing ratios with increasing latitude, which is most pronounced in hemispheric winter. Increased mixing ratios almost reach the tropopause for this season. The tropospheric distributions show a clear seasonality in both hemispheres with largest values observed in hemispheric winter and smallest values in hemispheric summer in the lower troposphere. There is a slight asymmetry towards generally higher mixing ratios in the NH, particularly for the 70° N bin. The NH has more coastal regions, which are assumed to be one of the main

sources of Br-VSLS, which could explain the asymmetry of tropospheric mixing ratios. A rather striking difference between the NH and SH is observed in the LMS in autumn and spring. While the distribution of $CH_2Br_2$ in hemispheric spring is quite similar, the distribution in hemispheric autumn differs with smaller values in the SH compared to the NH. High mixing ratios of tropospheric tracers in the LMS observations during NH autumn have been explained by strong influx of tropospheric air during NH summer and autumn ("flushing" of the LMS) (e.g., Hoor et al., 2005; Bönisch et al., 2009). It could be argued that

this is an indication of the different strength of tropospheric air mixing into the LMS of the two hemispheres. The subtropical jet acts as a transport barrier and Konopka et al. (2015) diagnosed a hemispheric asymmetry of the subtropical jet with a most pronounced weakening in the NH summer (see Fig. 1 therein).

    Even though SH observations are available for all seasons from the different missions, the SH database remains much smaller than the NH database. Unfortunately, MAM measurements during ATom–4 show quite large differences between the results

from TOGA and both Whole Air Sampler (WAS and PFP), but only for observations of the SH LMS. Fig. 4 displays the altitude-latitude cross section of the observations for MAM, taking all observations (a) and thus similar to Fig. 2 (b), as well as using all observations but only TOGA measurements from ATom (b) and using all observations but only the Whole Air Samplers for ATom (c). Although there is little difference in the rest of the atmosphere due to the use of TOGA or WAS/PFP, the difference in the SH LMS during MAM is clear (see Fig. 4 observations inside red rectangles). Using the TOGA data (Fig.

4 b), values are larger in the range of around 0.4–0.8 ppt, which would suggest a "flushing" of the LMS similar as in northern hemispheric autumn (e.g., Bönisch et al., 2009). Values are much smaller when using WAS and PFP data (Fig. 4 c) below around 0.4 ppt. This would indicate a strong isolation of the SH lower stratosphere. Indeed, Shuckburgh et al. (2009) investigated a strong seasonal cycle in the strength of the barriers at the subtropical jet, where in MAM in the SH, mixing follows a mostly zonal pattern and the subtropical jet acts as a barrier. They further stated that observed mixing is of greatest magnitude in the

NH in any season. The representation of the southern hemispheric UTLS in MAM is based on less observations than e.g., southern hemispheric UTLS in spring where the SouthTRAC campaign took place. The SouthTRAC campaign contributes to a substantial portion of the spring UTLS observations in the SH. For a more meaningful result especially in SH autumn, but also in winter and summer, further measurements are necessary and should be a focus for future campaigns.

    The model results show a general good agreement to the observations in the annual mean. Positive or negative bias to the

observations are nor very pronounced or consistent. Instead, negative or positive bias is dependent on season and latitude. For the case of $CH_2Br_2$, largest lower tropospheric values in the NH observations are in winter, whereas models show largest values in spring. This may arise from a possible incorrect seasonal representation in the Ordóñez et al. (2012) emission scenario.





Furthermore, TOMCAT values in the free troposphere are larger than in CAM-Chem, despite using the same emission scenario. Thus, overestimation in MAM in the NH is larger within TOMCAT with differences to observations up to about 0.4 ppt. Both

models underestimate NH high latitude values of $CH_2Br_2$ in SON by up to 0.3 ppt. Hossaini et al. (2016) showed a comparison of different models and ground based stations in which the models do not reproduce the seasonality at coastal stations such as Mace Head (Ireland) (see Fig. 3 therein). The observations used in this work were conducted predominantly over ocean and coastal regions (e.g., nearshore bases such as during the WISE campaign) and may have similar signatures to the coastal stations shown in Hossaini et al. (2016). Neither model shows the pronounced seasonal pattern of $CH_2Br_2$ in the SH with

elevated values in the southern hemispheric winter. This may be due to the emission scenario of Ordóñez et al. (2012), which was used in both models. The top-down emission estimates of the bromocarbons in the Ordóñez et al. (2012) scenario are based on aircraft campaigns and some available observations in the marine boundary layer, most of which are from the NH. Ordóñez et al. (2012) already identified some issues regarding the emission flux estimates in the SH as a consequence of missing aircraft observations in the SH. In addition, the Ordóñez et al. (2012) emission inventory does not consider Br-VSLS arsing from sea

ice regions, which are possible sources of Br-VSLS according to e.g., Abrahamsson et al. (2018).

As noted before, $CHBr_3$ as a much shorter atmospheric lifetime than $CH_2Br_2$. Consequently, the global tropospheric distributions from the observations of $CHBr_3$ show larger variability and a less pronounced seasonality. Jia et al. (2019) used simulations to demonstrate that uniform background emissions of $CHBr_3$ from the ocean result in a highly variable distribution in the atmosphere with larger values in regions of convergence or low wind speed and that the impact of localized elevated emissions

on the distribution varies significantly from campaign to campaign. Thus, the interpretation of $CHBr_3$ airborne observations is challenging. Nonetheless, the NH shows larger values in autumn and winter compared to spring and summer, a feature that is captured by both models, although the modeled wintertime maxima are more pronounced than in the observations. Based on the limited existing observations in the SH, the seasonality is not very pronounced with slightly smaller values in summer high latitudes and slightly larger near-ground values in autumn followed by winter. SH high latitude ground-based NOAA/GML

measurements presented in Hossaini et al. (2016) like the one from Cape Grim (Tasmania, Australia) and the Palmer Station (Antarctica), which are on consistent scale as aircraft measurements e.g., from PFP, also show a less pronounced seasonality. Furthermore, measurements of these stations reveal that models did not capture the observed seasonality for coastal stations (see Fig. 3 in Hossaini et al. (2016)). Observations and model comparison from this study demonstrate a similar discrepancy as seen in Hossaini et al. (2016) at high latitudes of the SH with an underestimation in summer and autumn, where differences

for tropospheric observations range between about 0.2 and 0.6 ppt, and overestimation mainly in winter high latitudes which reaches up to 0.8 ppt. Both models overestimate NH tropospheric values in winter and spring by up to 0.8 ppt except for near-ground values at high latitudes, which are much larger in the observational data (2–5 ppt difference to the observations above 80° N). In summer and autumn, the models underestimate high latitude values (differences to observations about 0.2 ppt and up to 0.9 ppt near the ground). Especially in the NH there is a larger frequency of observations over the northern Atlantic Ocean

(see Fig. 1). For example, the flights of the WISE campaigns took place predominantly from Shannon (Ireland), close to known coastline source regions of $CHBr_3$, which may have a greater influence on the atmospheric distribution than photochemical decay (e.g., Carpenter et al., 2005; Hossaini et al., 2016). $CHBr_3$ lower stratospheric values are close to zero, as this substance





has a shorter lifetime and is faster decomposed compared to $CH_2Br_2$. Model estimates of TOMCAT and CAM-Chem overall agree well within the LMS of both hemispheres.

## 4.2 Upper tropospheric latitudinal distribution

Trace gases can enter the extratropical UTLS through several pathways. Beside the downward transport from the stratospheric overworld, there is a two-way exchange across the extratropical tropopause and an isentropic exchange, often in the vicinity of the subtropical jet (e.g., Gettelman et al., 2011, and references therein). The amount of total bromine likely differs depending on how and where air enters the LMS and, consequently, the characteristics of the contributing input regions.

To investigate the distributions of the two major Br-VSLS in the upper troposphere as a function of latitude, we binned the observational data according to latitude and to potential temperature relative to the local tropopause. Only data in the 10 K range below the local dynamical tropopause are included, to characterize the possible input region. For the models, only data in the 10 K range below the climatological PV-based tropopause are included. Data have been separated into low latitudes (0°–30°), mid latitudes (30°–60°), and high latitudes (60°–90°) for both hemispheres. Results for $CH_2Br_2$ and $CHBr_3$ are listed in Tables 1 and 2, respectively, and the hemispheric winter $CH_2Br_2$ results are shown in Fig. 5 (graphical representations of the other seasons and seasonal cycle of each latitudinal band can be found in the supplement). Engel and Rigby (2018) reported typical tropical tropopause mixing ratios compiled from different measurement campaigns with mixing ratios in the upper tropical tropopause layer (upper TTL) of 0.73 (0.43–0.94) ppt $CH_2Br_2$ and 0.28 (0.02–0.64) ppt $CHBr_3$.

As shown in Table 1, low latitude $CH_2Br_2$ values on both hemispheres during all seasons are well within the range reported by Engel and Rigby (2018) although with slightly higher mixing ratios. Keber et al. (2020) showed a clear tendency for an increase in tropopause mixing ratios with latitude in the NH, most pronounced in winter (see Fig 5). This is likely due to the increase in lifetimes with latitude, as photochemical breakdown becomes slower with higher latitudes especially during winter. Our extended data set confirms the findings of Keber et al. (2020). In addition, we show that the SH upper tropospheric distribution looks similar to that of the NH. Mixing ratios in NH winter and spring are slightly larger than in SH winter and spring, whereas mixing ratios in the respective hemispheric summer are close to each other across all latitudes. Larger differences are observed in the low and mid latitudes of NH and SH during autumn, with larger values in NH than in SH. However, the high latitudes of each autumn differ only to a small extent.

Both models qualitatively reproduce the larger $CH_2Br_2$ values in hemispheric winter and spring and smaller values in summer and autumn (see Fig. S3). However, both models overestimate low latitude values for all seasons except for SH spring and NH winter. This overestimation in TOMCAT could already be seen in Keber et al. (2020) for the NH low latitudes for both winter and late summer to early autumn, as well as in Hossaini et al. (2013) where the TOMCAT model was only compared with HIPPO data. In general, CAM-Chem values are slightly smaller than TOMCAT values except in SH summer and autumn high latitudes. In NH winter, the models are close to observations (observations in between the model results), but both models overestimate NH spring values. However, both models reproduce the observed increase in mixing ratios with latitude in NH winter and spring, with CAM-Chem showing slightly better agreement. The models simulate SH winter and spring well in the mid and high-latitudes, with TOMCAT somewhat closer to the winter observations, but none of the models capture the





observed latitudinal variations in winter. Largest values in the models are found at high latitudes in spring for both hemispheres where observations show largest values in hemispheric winter high latitudes (see Table 1 for absolute values). Summer and autumn of both hemispheres are in better agreement with CAM-Chem, as TOMCAT deviates more from the observations.

TOMCAT simulates generally smaller mixing ratios in the SH and larger mixing ratios in NH summer and autumn mid and high latitudes.

For $CHBr_3$ (see Table 2 and Fig. S2 and S4), the upper tropospheric distribution is much more variable and shows a less clear seasonality, especially in the SH. NH low latitude values are slightly higher with a maximum in winter of 0.43 (0.33-0.62) ppt and maximum value in SH spring of 0.41 (0.36-0.43) ppt. NH values show an increase in mixing ratios with latitude,

most pronounced in winter and spring. In NH summer, mixing ratios drop from low to mid latitudes and increase towards high latitudes again. Thus, NH upper tropospheric distribution of $CHBr_3$ does not show the behavior described in Keber et al. (2020) with an increase towards mid latitudes and decrease at higher latitudes. Keber et al. (2020) considered a combined data set for summer and early autumn, whereas in this analysis summer and autumn are considered separately. As the latitudinal distribution differs substantially in summer and autumn (see Table 2), the separate consideration of summer and autumn compared to the

combined consideration as in Keber et al. (2020) can lead to differences in the interpretation of the observations. The behavior of $CHBr_3$, which can be seen in the NH summer, is also observed in the SH spring as well. All other seasons of the SH show an increase in mixing ratios with latitude, but with largest values in summer and autumn high latitudes with up to 0.57 (0.32-0.82) ppt in SH summer.

TOMCAT simulates an increase in mixing ratio with latitude for almost all seasons, with the largest values occurring in

hemispheric winter. Only in hemispheric summer, values decrease from low to mid latitudes and then increase slightly in SH high latitudes and remain at the same level in the NH high latitudes. Mid and high latitude mixing ratios in the NH are highly overestimated in winter, spring, and to a smaller extent in autumn, a feature already observed in Fig. 3. In the SH, mid latitude mixing ratios are also overestimated in winter, spring, and autumn which is also true for high latitude spring and autumn. In contrast, the observed and modeled mixing ratios in the SH high latitudes in winter and summer show a different behaviour. In

the observations, the high latitude values are smallest in the winter and highest in the summer, which is a reversed behaviour of the model estimations (see Table 2 or Fig. S4 in the supplement). As the high latitude observations are much more limited during these seasons (cf. Fig. 2 or Table S1 in the supplement), we are careful about interpreting these differences and further observations are needed to verify these deviations from model results. CAM-Chem upper tropospheric distributions are similar to TOMCAT distributions, although in general show smaller values for all latitudes. Thus, $CHBr_3$ mixing ratios from CAM-

Chem are somewhat closer to the observation in NH winter, spring, and autumn mid to high latitudes, and deviate slightly more in NH summer, compared to TOMCAT. In the SH, spring and autumn distributions from CAM-Chem are closer to observations compared to winter and summer distributions. Like for TOMCAT, the seasonal variation within the high latitudes show a different behaviour in comparison to the observations, with an even smaller value from CAM-Chem in SH summer high latitudes. Overall, the distribution of $CHBr_3$ is highly variable, and both models simulate similar latitudinal distribution, though

with smaller values for CAM-Chem. The tug-of-war between rapid advective transport and local accumulation at the time of





emission plays a decisive role. As already mentioned, Jia et al. (2019) showed that transport variations in the atmosphere itself produce a highly variable Br-VSLS distribution with elevated values not always reflecting strong localized sources.

### 4.3 Mid latitude UTLS Vertical profiles

The observational coverage of the upper troposphere and especially the lower stratosphere is best in spring and autumn of the
respective hemisphere. From the altitude-latitude cross sections (Sect. 4.1 Figs. 2 and 3), we already suspect larger differences of LMS Br-VSLS distribution in hemispheric autumn than in hemispheric spring.

We thus took a closer look at the vertical profiles of $CH_2Br_2$ and $CHBr_3$ in the mid latitudes of the SH and NH during these seasons. The observations from the different missions were seasonally combined and have been binned in 10 K intervals of potential temperature and potential temperature difference to the local tropopause. Only bins with at least five observations
are considered. For the profile in the free troposphere, data were binned in potential temperature ($\Theta$). Data binned in potential temperature difference to the local tropopause ($\Delta\Theta$) show larger variability in the free troposphere (e.g., Keber et al., 2020). $\Delta\Theta$–coordinates are therefore not well suited for tropospheric data. As $\Delta\Theta$–coordinates reduce the variability of the profile near the tropopause and within the lowermost stratosphere (e.g., Keber et al., 2020), the profile continued from 10 K below the local tropopause into the stratosphere in $\Delta\Theta$–coordinates. The two vertical coordinates were combined by aligning 0 K of
$\Delta\Theta$ with the median tropopause in $\Theta$, observed during the measurements. Profiles in $\Theta$–coordinates may extend beyond the median tropopause even if the observations are declared as tropospheric ones. Tropopause $\Theta$ of these observations are much larger and correspond to a higher tropopause, indicating that these observations may be subtropical in origin. The mixing ratios are averaged over equivalent latitude* of 40-60° of the respective hemisphere using box-and-whisker plots for the binned data (see Figs. 6 and 7). Vertical gradients for spring and autumn profiles for both hemispheres from tropopause values up to 30 K
above the local tropopause are summarized in Table 3.

Figure 6 shows hemispheric spring profiles of $CH_2Br_2$ and $CHBr_3$. Tropospheric values of $CH_2Br_2$ are very similar and close to 1 ppt in both hemispheres between 280 and 290 K of $\Theta$. NH tropopause values are 0.91 and 0.95 ppt using $\Theta$ and $\Delta\Theta$ as vertical coordinates and SH tropopause values are 0.86 and 0.85 ppt using $\Theta$ and $\Delta\Theta$ and thus only slightly smaller. The $CH_2Br_2$ profiles in the lowermost stratosphere of both hemispheres are very similar in their respective spring with lowest values
of around 0.38 to 0.39 ppt in the NH and 0.37 to 0.39 ppt in the SH below 90 K $\Delta\Theta$ beside an exceptional low value of 0.25 ppt at 90 K of $\Delta\Theta$ in the NH. Vertical gradients of $CHBr_3$ profiles are larger compared to $CH_2Br_2$, well in line with the much shorter lifetimes (see Table 3). Tropospheric mixing ratios are 0.81 ppt between 280 and 290 K $\Theta$ in NH mid latitudes, while SH mid latitudes mixing ratios are slightly larger with 0.96 ppt between 280 and 290 K $\Theta$. However note that NH tropospheric values present a larger variability. In contrast, tropopause values in the NH are slightly larger with 0.6 and 0.57 ppt using $\Theta$ and
$\Delta\Theta$ compared to 0.43 in both $\Theta$ and $\Delta\Theta$ in the SH. Mixing ratios drop to values clode to zero on both hemispheres at about 30–40 K of $\Theta$ above the tropopause.

Figure 7 shows that hemispheric autumn profiles of $CH_2Br_2$ and $CHBr_3$ are less similar than in hemispheric spring. Tropospheric values of $CH_2Br_2$ are 0.99 and 0.83 ppt for NH and SH between 280 and 290 K of $\Theta$ and thus slightly larger in the NH. Tropopause values are 0.77 and 0.78 ppt in the NH compared to 0.72 and 0.73 ppt in the SH using $\Theta$ and $\Delta\Theta$ as the vertical





coordinate. Thus, tropopause values are slightly smaller compared to spring values but hemispheric differences in spring and autumn are comparable. Differences between the hemispheres become larger on lowest levels above the dynamical tropopause, i.e. in the ExTL. $CH_2Br_2$ shows a larger vertical gradient up to 30 K of $\Delta\Theta$ in SH autumn than in NH autumn (see Table 3) reaching smallest value of 0.45 ppt between 20 and 30 K of $\Delta\Theta$. With a comparable distance to the dynamic tropopause, the value in the NH is 0.64 ppt. Between 40 and 70 K of $\Delta\Theta$, SH values range from 0.53 to 0.7 ppt associated with a large

variability.

    Figure 8 shows again the SH autumn vertical profile of $CH_2Br_2$ but two more profiles were included in $\Delta\Theta$ coordinates with one profile excluding TOGA observations and one profile excluding WAS and PFP observations. The profile including only TOGA from ATom is much closer to the profile using all observations, whereas the profile including only WAS and PFP observations from ATom shows a much steeper gradient across the tropopause and much smaller values. The larger amount of

TOGA observations shifts the median towards larger values. The profile without TOGA measurements would be in line with the assumption of a strong transport barrier, e.g., the subtropical jet and exchange between stratosphere and troposphere only at the edges of the jets (e.g., Fig. 1 in Gettelman et al., 2011), while the profile without WAS and PFP shows a less strong but still larger transport barrier than in NH autumn profile (Fig. 7 b). The NH profile also shows a much smaller gradient compared to NH spring and less variability which may indicate a well mixed LMS. This is also well in line with finding by e.g., Bönisch

et al. (2009) who showed a "flushing" of the lowermost stratosphere in summer and autumn.

    Hemispherical differences can also be inferred from $CHBr_3$ vertical profiles. Larger differences can be seen in tropospheric values between 280 and 290 K $\Theta$ with 1.76 ppt in the NH and 0.87 ppt in the SH. In contrast, hemispheric differences at the tropopause are small with 0.4 and 0.43 ppt (NH) compared to 0.49 and 0.43 ppt (SH) using $\Theta$ and $\Delta\Theta$ as vertical coordinates. Regarding the LMS and especially the ExTL, the vertical gradient is much larger in the SH than in the NH. This implies that

there is a stronger transport barrier in the SH when looking at $CHBr_3$. SH values drop down to 0.05 ppt between 10 and 20 K above the dynamical tropopause, much lower than 0.22 ppt at comparable distance to the NH autumn tropopause. Furthermore, only two more values around 0.02 ppt can be assigned between 50 and 70 K of $\Delta\Theta$, whereas NH lowermost stratospheric values are above 0.02 ppt even at 90 K of $\Delta\Theta$.

    We further compared the mid latitude profiles of $CH_2Br_2$ and $CHBr_3$ with model results of TOMCAT and CAM-Chem in

Figs. 9 and 10. The comparisons are shown as a function of $\Delta\Theta$. Since there is no tropopause information for the TOMCAT model, we derived $\Delta\Theta$ as the difference of the climatological tropopause potential temperature and model potential temperature. For consistency, the climatological tropopause was used for CAM-Chem as well. Additionally, since equivalent latitude information is not available for the models, latitude was used instead. The mean absolute percentage differences (MAPDS) reported below are tabled in the supplements.

For spring $CH_2Br_2$ (Fig. 9 a–b), TOMCAT profiles overestimate the observations on average by about 0.12 ppt in the NH and 0.08 ppt in the SH, with a corresponding MAPD of about 26% (NH) and 18% (SH). The CAM-Chem profiles are closer to the observations. Particularly the NH profile within the lowest 20 K above the dynamical tropopause is close to the observations but deviates towards higher altitudes. Average differences in the NH are 0.06 ppt and a MAPD of 17%, whereas in the SH, differences are on average 0.03 ppt with a MAPD of 9%. For $CHBr_3$ (Fig. 9 c–d), the comparison is only shown up to 40 K of





$\Delta\Theta$, since this substance was almost completely depleted above. TOMCAT seems to overestimate NH CHBr$_3$ by on average 0.1 ppt, corresponding to a MAPD of 69%. Differences are much smaller in the SH with 0.02 ppt and a MAPD of 21%. CAM-Chem estimated well the lowermost stratosphere in the NH with an average difference of 0.02 ppt and a MAPD of 15%. SH differences of CAM-Chem are on average 0.04 ppt with a MAPD of 26%, thus slightly larger than TOMCAT differences. Both models estimated a nearly complete depletion of CHBr$_3$ in the SH lower stratosphere (above 40 K of $\Delta\Theta$). However,

observations show a slight offset to the models estimations with values of 0.02–0.04 ppt up to 90 K of $\Delta\Theta$ (Fig. 9 c).

Hemispheric autumn mid latitude profiles of CH$_2$Br$_2$ and CHBr$_3$ are displayed in Fig. 10. In NH autumn, the observed profiles and those of the models agree well. In the SH, however, the profiles diverge further as neither of the models reproduce the presumably stronger barrier at the tropopause described in Fig. 8. TOMCAT overestimates NH CH$_2$Br$_2$ up to 90 K of $\Delta\Theta$ on average by about 0.1 ppt (MAPD of 20%). The difference between models and observations becomes much larger in

SH autumn. In addition, the observational profile exhibits a high degree of variability, and the scatter of the individual bins increases due to the differences between the ATom instruments, as mentioned earlier. Although models are close to observations above 40 K of $\Delta\Theta$, average differences between 0 and 40 K of $\Delta\Theta$ are 0.12 ppt (MAPD of 25%). CAM-Chem estimates are closer to the observations in the NH and differs on average by about 0.02 ppt corresponding to a MAPD of 6%. However, CAM-Chem showed a similar profile like TOMCAT in the SH, although slightly smaller values in general and thus a smaller

average difference between 0 and 40 K of $\Delta\Theta$ of 0.06 ppt and a MAPD of 15%. Compared to NH spring, TOMCAT shows a CHBr$_3$ profile which is much closer to the observations in NH autumn. The average difference below 40 K of $\Delta\Theta$ is 0.06 ppt with a corresponding MAPD of 22%. SH observations are limited and thus a comparison is only possible up to 20 K of $\Delta\Theta$. Although observations and model results of TOMCAT are close near the dynamical tropopause, the difference increases rapidly to 0.13 ppt between 10 and 20 K of $\Delta\Theta$. As the CAM-Chem profile is smaller in absolute values, the largest differences are

around 0.08 ppt. Both models fail to capture the steeper gradient across the dynamical tropopause during SH autumn.

## 5 Summary and conclusion

In the present work, we investigated the global seasonal distribution of the two major short-lived brominated substances CH$_2$Br$_2$ and CHBr$_3$. These natural substances with dominant oceanic origin gain importance because their relative contribution to the loss of ozone will rise as a result of the decline of the long-lived brominated substances of anthropogenic origin. We used data

from four HALO missions: TACTS, WISE, PGS, and SouthTRAC. To further expand the data set, we included aircraft observations of the HIPPO and ATom missions. These are two global scale missions, covering a wide latitude range in all seasons from the ground to the lowermost stratosphere. Zonal mean distributions were analyzed by using latitude as the horizontal and pressure as the vertical coordinate (altitude-latitude cross sections). As the focus moved on to the tropopause region and lowermost stratosphere (upper tropospheric distribution and vertical profiles), data are presented in potential temperature and

further in a tropopause relative coordinate namely the difference in potential temperature to the dynamical tropopause ($\Delta\Theta$). We further compared the observed distributions with two model distributions from TOMCAT and CAM-Chem, both using the Ordóñez et al. (2012) emission scenario, with emissions varying by season.


We found a similar tropospheric seasonality of $CH_2Br_2$ in both hemispheres, although with slightly larger mixing ratios in the NH. The larger values in the NH agree with expected hemispheric difference, as the main sources of many brominated

VSLS are believed to be stronger from coastal ocean regions. The ratio of ocean and land mass of the NH and SH is different, causing the size of coastal areas to vary. The upper tropospheric distributions of $CH_2Br_2$ also show that the seasonality is similar in both hemispheres with larger values in winter and spring and smaller values in summer and autumn. In addition, in all seasons the mixing ratios are larger at NH mid and high latitudes than at the SH mid and high latitudes. Global seasonal distribution of $CHBr_3$ shows larger variability and less clear seasonality. Although NH mixing ratios seem to be larger in winter

and spring than in summer and autumn, SH distributions show less seasonality with slightly larger values in autumn but overall smaller than in the NH. This may again be a result of the different ratio of ocean and land mass of both hemispheres. In good agreement with Keber et al. (2020), the mixing ratios at the extratropical tropopause are systematically larger than those at the tropical tropopause in both hemispheres at all times of the year. The comparison of the distributions in the UTLS was limited to hemispheric spring and autumn due to a lower coverage of the SH by observations. Mid latitude profiles of $CH_2Br_2$ and $CHBr_3$,

extending into the lowest stratosphere in hemispheric spring, are similar. Profiles in hemispheric autumn differ much more. In particular, SH profiles of $CH_2Br_2$ and $CHBr_3$ show much steeper gradients across the tropopause and into the stratosphere than NH profiles, and in the case of $CH_2Br_2$, SH observations present higher profile variability. This provides room for discussion as to whether the transport barrier in the SH autumn is significantly stronger, preventing the "flushing" of the lower stratosphere as it occurs in the NH summer and autumn. Unfortunately, in particular the observations in the lower stratosphere of the SH

autumn show large differences between the different instruments on board the aircraft during ATom–4. Even if differences in the hemispheric autumn are already recognizable here, more observations and further investigation are needed to confirm an interhemispheric difference in the respective autumn lowermost stratosphere. The SH is less sampled than the NH and the observations may not be representative of the general distribution for all seasons in the SH.

We further compared the observed and modelled distributions from TOMCAT and CAM-Chem. The observed seasonality

of $CH_2Br_2$ was only partially reproduced by the models and was not very pronounced in the SH, probably due to the used emission scenario, which for the high latitudes present a homogeneous distribution and less pronounced seasonal cycle due to the scarcity of observations used in the construction of the scenario. The lack of aircraft observations in the SH led to issues regarding the emission flux estimates. In the case of $CHBr_3$, both models systematically overestimate hemispheric winter and spring mixing ratios in the free troposphere and slightly underestimate hemispheric summer and autumn. In general, the

mixing ratios in TOMCAT are larger than in CAM-Chem, which could be due to the differences in transport and the efficiency of the photochemical decomposition in the models when using the same emission scenario. Regarding upper tropospheric distributions, the models reproduced well the seasonality with larger values in winter and spring and smaller values in summer and autumn. Model estimates are close to $CH_2Br_2$ observations for all seasons and both hemispheres, although the different "flushing" behavior in the lower stratosphere between NH and SH is not captured by any of the models. Regarding $CHBr_3$,

both models yield significantly higher values in hemispheric winter mid and high latitudes than observed. The high latitude observations of $CHBr_3$ of the SH show a strong deviation from the models. Especially in winter and summer, the deviations are particularly large with observations showing reversed behavior to the model simulations (largest values in summer and





smallest values in winter). At these times of the year the flights do not reach such high latitudes ($< 70°$ S) and the number of observations in the high latitudes is very limited. For a more meaningful conclusion for the SH high latitudes, additional

observations are needed. Modelled mid latitude vertical profiles agree well with observed profiles for NH spring and autumn as well as for SH spring. TOMCAT profiles are always slightly higher in mixing ratios than CAM-Chem and differences were therefore predominantly smaller when comparing with CAM-Chem. Both models were not capable to reproduce SH autumn vertical profiles of $CH_2Br_2$. Vertical gradients of $CH_2Br_2$ and $CHBr_3$ across the tropopause in the respective autumn differ strongly from each other with steeper gradients in the SH autumn. Vertical gradients in the hemispheric spring, on the other

hand, are more similar to each other.

Given these initial results of the global distribution of the two major Br-VSLS, we reinforce the utility and need for further observations in the SH UTLS to further understand the seasonal distribution of theses species. Especially in the southern hemispheric UTLS, data coverage remains sparse in most seasons. First differences of the NH and SH could already be indicated based on the data used, thus extrapolating northern hemispheric observations to the SH is not possible. The representation of

seasonal variability of Br-VSLS emissions and the efficiency of photochemical processes within the high latitudes need to be improved individually for the NH and SH to improve the agreement with current and future observations. In addition, it is of importance to generate a long-term global dataset that can be used to determine if there is a trend in Br-VSLS abundance at the global and hemispheric scale.

*Data availability.* Observational data from the HALO missions are available via the HALO Database (halo-db.pa.op.dlr.de/). ATom obser-

vational data are available at the Oak Ridge National Laboratory Distributed Active Archive Center (ORNL DAAC; Wofsy et al. (2021)). HIPPO observational data are available at the Earth Observing Laboratory data archive (EOL data archive, Wofsy et al. (2017)). Model data are available on request: please contact Ryan Hossaini (r.hossaini@lancaster.ac.uk) for TOMCAT model data and Rafael P. Fernandez (rpfernandez@mendoza-conicet.gob.ar) for CAM-Chem model data.

*Author contributions.* MJ, TK, TS, TW, and AE were involved in developing and operating the GhOST instrument. ELA and SM operated

the instruments during HIPPO and ECA, RSH, SM, and DRB operated the instruments during ATom. RH, RPF, ASL, and J-UG have provided model data and participated in the discussion regarding model comparison. MJ wrote the article with involvement of AE and TS in the analysis and writing process. All co-authors were involved in the discussion and iteration process of the article.

*Competing interests.* The authors declare that they have no conflict of interest.

*Acknowledgements.* This research has been supported by the Bundesministerium für Bildung und Forschung (grant no. 01LG1908B) as part

of the ROMIC II program and by the Deutsche Forschungsgemeinschaft as part of the HALO priority program (grant nos. EN367/13-1,



EN367/14-1, EN367/16-1, EN367/17-1) and as part of the collaborative research program "The Tropopause Region in a Changing Atmosphere" (TPChange, DFG TRR 301). RH was supported by a NERC Independent Research Fellowship (NE/N014375/1). Dough Kinnison for helping with the high-resolution SD-CAM-Chem setup. This material is based upon work supported by the National Center for Atmospheric Research, which is a major facility sponsored by the National Science Foundation under Cooperative Agreement No. 1852977. RPF would like to thank the financial support from CONICET and ANPCyT (PICT 2019-2187).





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



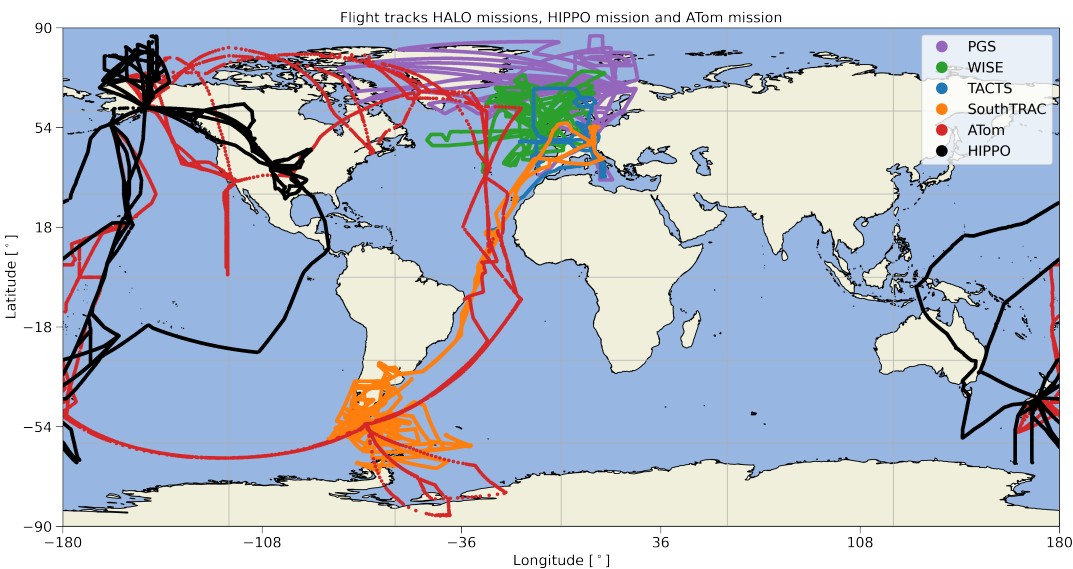

**Figure 1.** Flight tracks of the HALO missions TACTS (blue), WISE (green), PGS (purple), and SouthTRAC (orange) as well as flight tracks of the HIPPO mission (black) and ATom mission (red).



**Figure 2.** Altitude-latitude cross section of CH$_2$Br$_2$. The data are separated by season and displayed as a function of latitude and pressure. Top row (a–d) shows observational data. Second and third row (e–h and i–l) show model results of TOMCAT and CAM-Chem, respectively. Fourth and fifths row (m–p and q–t) show differences of the respective model to the observations. The dynamical tropopause (dashed lines) has been derived from ERA-Interim reanalysis, providing a climatological (1988 - 2018) zonal mean tropopause.





**Figure 3.** As in Fig. 2 but for CHBr$_3$.

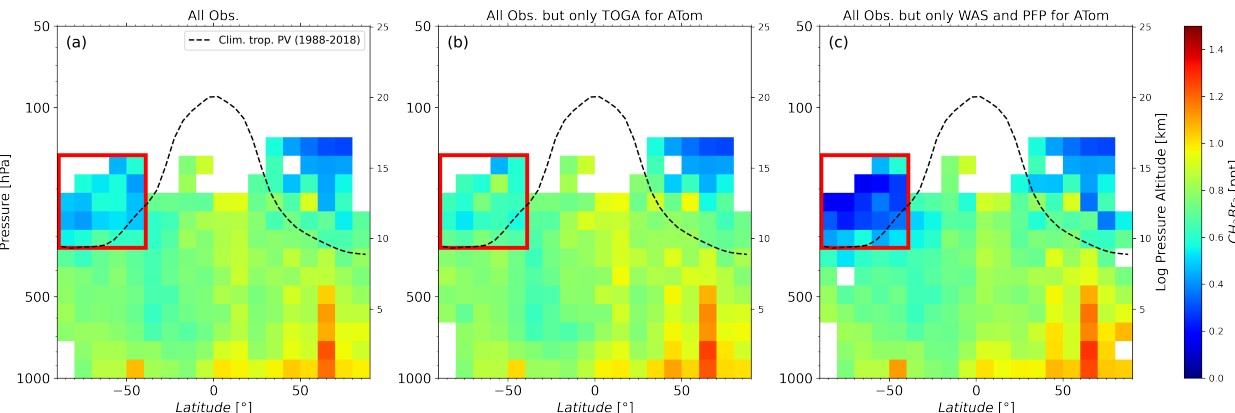

**Figure 4.** Altitude-latitude cross section of $CH_2Br_2$ for March, April, and May. (a) presents the distribution of $CH_2Br_2$ as in Fig. 2, whereas in (b) data from all missions were used, but only TOGA observations from ATom were included, and in (c) data from all missions were used, but only WAS and PFP observations from ATom were included. The red rectangle indicated the region where observations from different techniques differed substantially from one another.



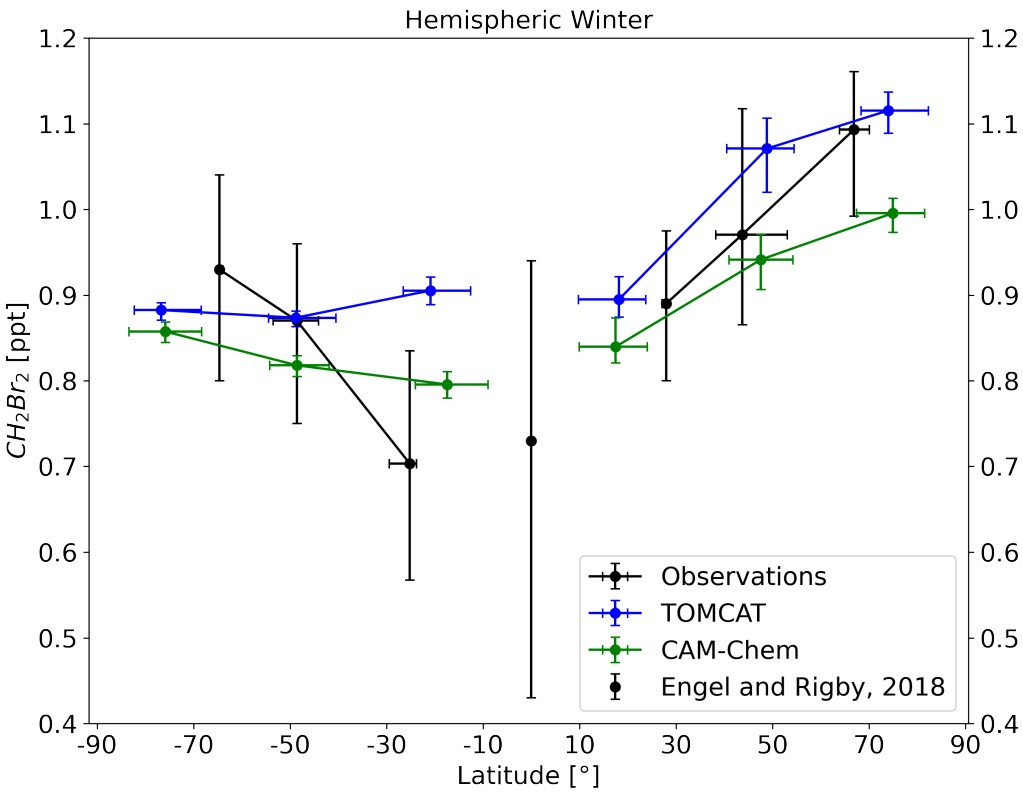

**Figure 5.** Latitude cross section of tropopause representative mixing ratios of $CH_2Br_2$ from observation and model results for both hemispheres in winter. Data are binned into three latitude bins for each hemisphere: high latitudes (60-90°), mid latitudes (30-60°), and low latitudes (0-30°) and only for data within the 10 K below the dynamical tropopause. Shown are the medians with the error bars representing the interquartile range (IQR). The median latitudinal position of observational and model bins may differ due to different spatial data coverage of observations and models. Also included is the reference mixing ratio for the tropical tropopause (Engel and Rigby, 2018).



**Figure 6.** Hemispheric spring vertical profiles of (a, b) $CH_2Br_2$ and (c, d) $CHBr_3$. Observations were averaged over 40–60° of equivalent latitude*. Data are displayed as a function of potential temperature for tropospheric values (black) and potential temperature difference to the local tropopause for values from 10 K below the tropopause and above (red). Shown are the medians with the boxes representing the interquartile range (IQR), whiskers as the 1.5×IQR and circles are single observations outside the whiskers (outliers). The dashed black line shows the median dynamical tropopause derived from the times and locations of the observation.



**Figure 7.** As in Fig. 6, but for hemispheric autumn.



**Figure 8.** As in Fig. 7, but only $CH_2Br_2$ for SH autumn. Profile as a function of potential temperature difference to the local tropopause was split into three profiles. Profile using all observations is in red, similar to Fig. 7, profile using all observation but with only TOGA observations from ATom in dark red and profile using all observations but with only flask observations in light red.



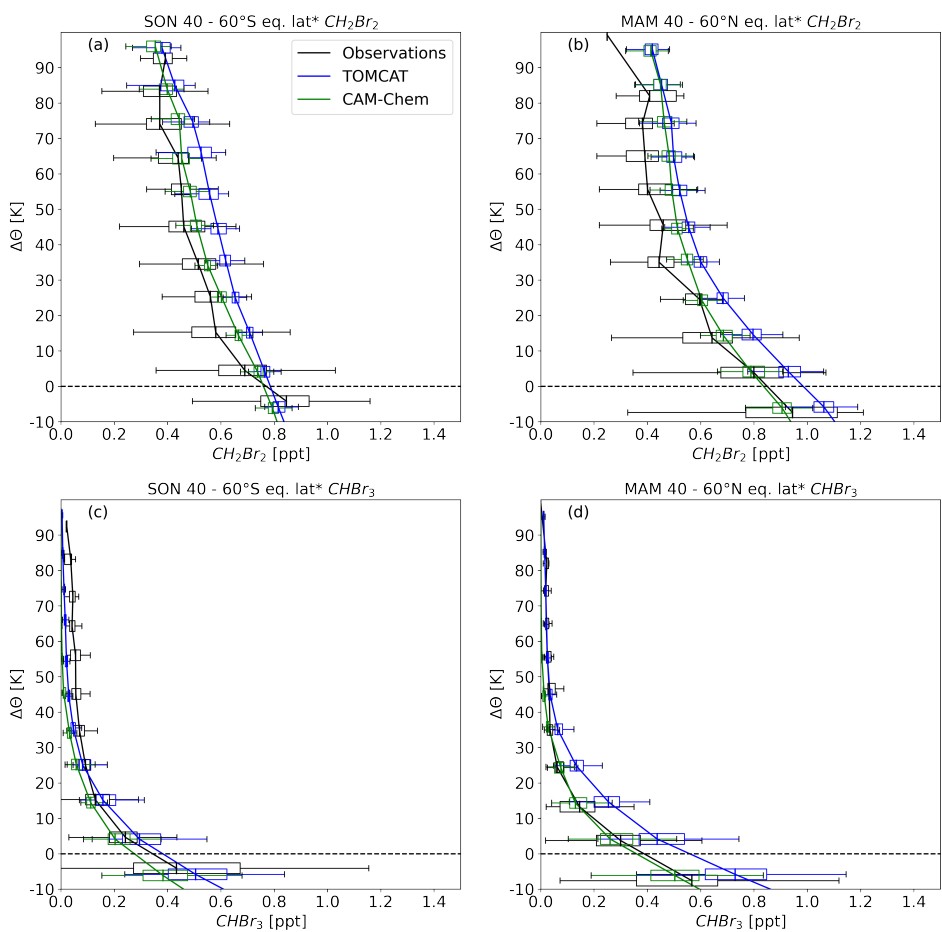

**Figure 9.** Hemispheric spring vertical profiles of (a,b) $CH_2Br_2$ and (c, d) $CHBr_3$. Observations were averaged over 40–60° of equivalent latitude[*]. Data are displayed as a function of potential temperature difference to the local tropopause (black). Also shown are model results from TOMCAT (blue) and CAM-Chem (green) as a function of potential temperature relative to the climatological tropopause. Profiles show medians with the the boxes representing the interquartile range (IQR) and whiskers as the 1.5×IQR. Outliers are not included for a better illustration.



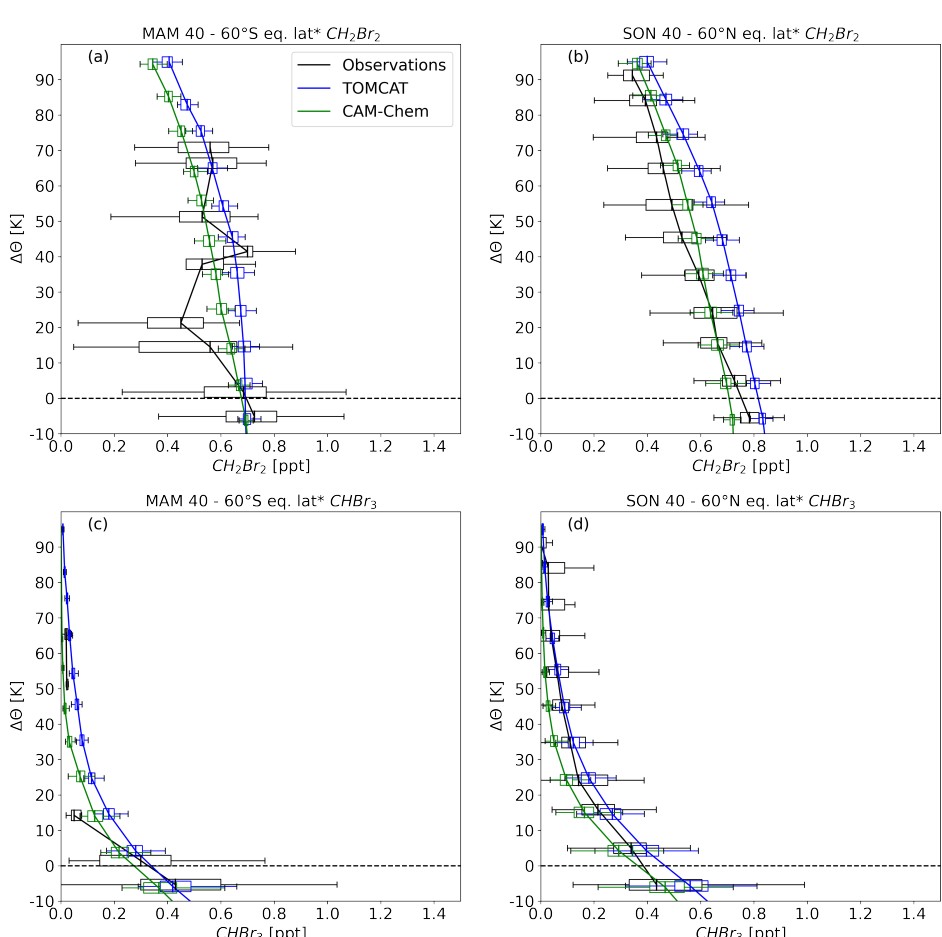

**Figure 10.** As in Fig. 9, but for hemispheric autumn.



**Table 1.** Averaged mole fractions (median in parts per trillion, ppt) of $CH_2Br_2$ and their corresponding range (25[th] to 75[th] percentiles) at high latitudes (60°-90°), mid latitudes (30°-60°) and low latitudes (0°-30°) in the upper troposphere, e.g., within 10 K below the local dynamical tropopause for the Northern and Southern Hemispheres.

| $CH_2Br_2$ | Southern Hemisphere | | | Northern Hemisphere | | |
|---|---|---|---|---|---|---|
| | high latitudes | mid latitudes | low latitudes | low latitudes | mid latitudes | high latitudes |
| Observations | ppt (range) | | | ppt (range) | | |
| Winter | 0.93 (0.8-1.04) | 0.87 (0.75-0.96) | 0.7 (0.57-0.84) | 0.89 (0.8-0.98) | 0.96 (0.87-1.11) | 1.1 (1.04-1.17) |
| Spring | 0.87 (0.78-0.96) | 0.81 (0.73-0.91) | 0.86 (0.76-0.87) | 0.74 (0.69-0.83) | 0.84 (0.72-1.07) | 0.92 (0.75-1.15) |
| Summer | 0.73 (0.68-0.86) | 0.68 (0.59-0.76) | 0.68 (0.61-0.76) | 0.64 (0.49-0.71) | 0.7 (0.62-0.79) | 0.72 (0.64-0.81) |
| Autumn | 0.76 (0.67-0.82) | 0.7 (0.6-0.8) | 0.65 (0.57-0.73) | 0.74 (0.72-0.77) | 0.78 (0.75-0.82) | 0.77 (0.71-0.84) |
| TOMCAT | | | | | | |
| Winter | 0.88 (0.87-0.89) | 0.87 (0.86-0.88) | 0.91 (0.89-0.92) | 0.89 (0.87-0.92) | 1.07 (1.02-1.11) | 1.12 (1.09-1.14) |
| Spring | 0.89 (0.87-0.91) | 0.81 (0.79-0.83) | 0.84 (0.82-0.87) | 0.9 (0.87-0.92) | 1.04 (0.99-1.09) | 1.16 (1.13-1.19) |
| Summer | 0.66 (0.65-0.67) | 0.65 (0.64-0.69) | 0.83 (0.8-0.86) | 0.89 (0.87-0.9) | 0.81 (0.8-0.83) | 0.82 (0.81-0.83) |
| Autumn | 0.69 (0.68-0.69) | 0.7 (0.69-0.74) | 0.86 (0.84-0.88) | 0.88 (0.86-0.91) | 0.84 (0.82-0.85) | 0.83 (0.82-0.84) |
| CAM-Chem | | | | | | |
| Winter | 0.86 (0.84-0.87) | 0.82 (0.8-0.83) | 0.8 (0.78-0.81) | 0.84 (0.82-0.87) | 0.94 (0.91-0.97) | 1.0 (0.97-1.01) |
| Spring | 0.87 (0.85-0.89) | 0.79 (0.77-0.81) | 0.79 (0.77-0.81) | 0.82 (0.8-0.83) | 0.89 (0.84-0.93) | 1.01 (0.98-1.03) |
| Summer | 0.73 (0.71-0.75) | 0.68 (0.67-0.7) | 0.79 (0.76-0.82) | 0.77 (0.74-0.79) | 0.68 (0.67-0.69) | 0.7 (0.68-0.71) |
| Autumn | 0.73 (0.71-0.74) | 0.7 (0.69-0.71) | 0.79 (0.77-0.81) | 0.81 (0.79-0.83) | 0.71 (0.71-0.74) | 0.72 (0.71-0.73) |





**Table 2.** As in Table 1, but for CHBr$_3$.

| CHBr$_3$ | Southern Hemisphere | | | Northern Hemisphere | | |
| --- | --- | --- | --- | --- | --- | --- |
| | high latitudes | mid latitudes | low latitudes | low latitudes | mid latitudes | high latitudes |
| Observations | ppt (range) | | | ppt (range) | | |
| Winter | 0.36 (0.2-0.57) | 0.35 (0.19-0.48) | 0.2 (0.07-0.29) | 0.43 (0.33-0.62) | 0.54 (0.42-0.75) | 0.72 (0.63-0.82) |
| Spring | 0.45 (0.35-0.56) | 0.33 (0.24-0.52) | 0.41 (0.36-0.43) | 0.35 (0.3-0.46) | 0.49 (0.33-0.62) | 0.5 (0.34-0.8) |
| Summer | 0.57 (0.32-0.82) | 0.3 (0.2-0.47) | 0.3 (0.19-0.34) | 0.32 (0.26-0.42) | 0.27 (0.19-0.42) | 0.37 (0.28-0.56) |
| Autumn | 0.53 (0.38-0.76) | 0.38 (0.26-0.57) | 0.22 (0.17-0.28) | 0.28 (0.2-0.33) | 0.44 (0.33-0.6) | 0.45 (0.35-0.63) |
| TOMCAT | | | | | | |
| Winter | 1.27 (1.17-1.34) | 0.87 (0.69-1.04) | 0.28 (0.23-0.33) | 0.3 (0.25-0.37) | 1.18 (0.91-1.39) | 1.56 (1.43-1.68) |
| Spring | 0.83 (0.75-0.89) | 0.45 (0.33-0.59) | 0.23 (0.19-0.28) | 0.25 (0.21-0.31) | 0.67 (0.52-0.82) | 1.08 (0.99-1.18) |
| Summer | 0.28 (0.24-0.3) | 0.21 (0.19-0.24) | 0.29 (0.24-0.39) | 0.41 (0.33-0.49) | 0.28 (0.25-0.31) | 0.28 (0.25-0.3) |
| Autumn | 0.64 (0.57-0.69) | 0.4 (0.35-0.46) | 0.29 (0.25-0.35) | 0.41 (0.32-0.5) | 0.55 (0.48-0.61) | 0.73 (0.66-0.79) |
| CAM-Chem | | | | | | |
| Winter | 1.27 (1.16-1.36) | 0.73 (0.56-0.9) | 0.12 (0.09-0.18) | 0.12 (0.09-0.21) | 0.95 (0.71-1.16) | 1.41 (1.27-1.53) |
| Spring | 0.7 (0.61-0.77) | 0.35 (0.26-0.45) | 0.1 (0.07-0.14) | 0.12 (0.09-0.15) | 0.46 (0.33-0.57) | 0.77 (0.7-0.86) |
| Summer | 0.19 (0.17-0.22) | 0.16 (0.14-0.18) | 0.13 (0.09-0.21) | 0.26 (0.17-0.33) | 0.2 (0.17-0.21) | 0.2 (0.19-0.21) |
| Autumn | 0.64 (0.56-0.7) | 0.33 (0.28-0.41) | 0.13 (0.09-0.18) | 0.22 (0.16-0.32) | 0.44 (0.37-0.52) | 0.66 (0.59-0.71) |



**Table 3.** Vertical gradients across the tropopause in the Northern Hemisphere (NH) and Southern Hemisphere (SH) spring and autumn from tropopause (TP) mixing ratios from the 10 K bin below the dynamical tropopause up to the 20–30 K bin above the dynamical tropopause. In addition, local lifetimes of $CH_2Br_2$ and $CHBr_3$ for the tropospheric tropics and northern hemispheric mid latitudes at 10 km, taken from Carpenter et al. (2014)

| | Gradients [%/K] | | | | Local Lifetimes (days) | | | | |
| | Spring | | Autumn | | Tropics | Mid latitudes | | | |
| | SH | NH | SH | NH | | Winter | Spring | Summer | Autumn |
| --- | --- | --- | --- | --- | --- | --- | --- | --- | --- |
| $CH_2Br_2$ | 1.12 | 1.23 | 1.27 | 0.59 | 150 | 890 | 360 | 150 | 405 |
| $CHBr_3$ | 2.63 | 2.97 | 4.42 [a] | 2.25 | 17 | 88 | 29 | 17 | 44 |

[a] Gradient from TP to (10–20) K