# Peer review of "Global seasonal distribution of CH2Br2 and CHBr3 in the upper troposphere and lower stratosphere"

_Atmospheric Chemistry and Physics, 2022_

## Author Comment (AC1)

**Global seasonal distribution of CH$_2$Br$_2$ and CHBr$_3$ in the upper troposphere and lower stratosphere**

We would like to thank both reviewers for their constructive and detailed comments on the manuscript.
In the following, we address the respective proposals for improvement. Changes are explained in detail, answering each referee point by point. Reviewer comments are in normal font. Our answers are in italic and changes to the manuscript in blue.

**Response to Referee #1**

Specific comments:

-   Line 31: There's an extra 'to' here, maybe remove the first one.

    *Done.*

-   Lines 49-50: Add 'tropospheric' after 'extratropical' here.

    *Done.*

-   Figure 4: This is important to show the uncertainty that can exist between data from different instruments and how it can defy easy explanation. It's unfortunate that the discrepancy is so large in the region and time of interest but this makes it even more important to point out as you have done.

-   Line 261: 'as' should be 'has'.

    *Done.*

-   Line 276: 'tropospheric' misspelled

    *Done.*

-   Line 380: 'close' misspelled

    *Done.*

-   Section 4.3: Starting at about line 370 I really had trouble staying focused while reading this section because there are far too many listing of exact ppt values for each species at various levels and seasons. This is in contrast to Sections 4.1 and 4.2 that were easy to follow and had many interesting features. I would suggest removing nearly all mention of exact mixing ratios in the text, the numbers are in the figures if anybody wants to see them, and stick with describing the main points you want to discuss.

*We agree that some restrictions need to be applied to the concrete numbers in section 4.3, especially to the numbers for the tropopause, where we had given mixing ratios in two vertical coordinates. To do this, we have shortened section 4.3 from the given point and avoided exact numbers in places that can lead to confusion.*
*For the section, where modeled and observational profiles are compared, we suggest leaving the absolute differences and the relative difference (in MAPD)*
*The changes to this section can be found in the marked-up manuscript, as the section with changes is too large to show in here.*

**Response to Referee #2**

General comments:

1. Please clarify what novel findings were made by integrating and analyzing data from the northern and southern hemispheres in the abstract and summary sections.

   *In the abstract, we have now integrated the new findings/data on the differences in the lowest stratosphere by showing the small difference in the hemispheric spring (with a rough value of the maximum difference) and for hemispheric autumn. For autumn we do not give a rough value, because this difference is still associated with a high uncertainty, which is briefly explained in the following sentences in the abstract.*

   "[…] The lowermost stratosphere of SH and NH show a very similar distribution of $CH_2Br_2$ in hemispheric spring with differences well below 0.1 ppt,
   while the differences in hemispheric autumn are much larger with substantially smaller values in the SH than in the NH.
   […]
   The observations of $CHBr_3$ support the suggestion, with a steeper vertical gradient in the upper troposphere and lower stratosphere in SH autumn than in NH autumn. […]"

   *In addition, we included this information to the summary.*

   "[…] The LMS distributions in hemispheric spring are very similar (with differences well below 0.1 ppt) but differ considerably in hemispheric autumn (up to 0.3 ppt more in the NH). Mid latitude profiles of $CH_2Br_2$ and $CHBr_3$, extending into the lowest stratosphere in hemispheric spring, are also similar, whereas profiles in hemispheric autumn differ much more. […]"

2. Can the limited temporal and spatial sampling, especially in the southern hemisphere, affect the conclusion of this paper? However, there is no quantitative discussion on such effects; could they be assessed using TOMCAT and CAM-Chem model data?

*The limited number of observations does affect the conclusion and is also part of the summary and conclusion section. Not only for the lowermost stratospheric comparison but also the seasonality of SH high latitudes where observations are very limited in summer and winter. A quantitative investigation of, e.g., model data along the flight trajectories and a comparison with the zonally averaged model data would be one possibility but is beyond the scope of this manuscript.*

3. The methodology, especially the section on analytical methods, is insufficient. I would like additional explanation. Seems like different methods (e.g., with respect to equivalent latitude, sampling, etc.) were applied for observations and models, but I'm confused. Please sort them out.

   *This comment is related to a specific comment below. We have gone into more detail on the analytical method in that specific comment. A description on model data is now included in the newly formed section 4.1 "Analyses methods", as well as how equivalent latitude was derived in section 3.3. In each section of the results, we added additional information on sampling observations and models. Section 4.1 serves only as a brief overview of the analysis methods used in each results section.*

Specific comments:

- p. 1, l. 12: "the same emission inventory" What emission inventory?

  *We now specifically mention the emission inventory of Ordóñez et al. (2012) in the abstract.*
  *Thus, readers know directly which inventory the models work with.*

  "[...] We further compare the observations to model estimates of TOMCAT and CAM-Chem, both using the same emission inventory of Ordóñez et al. (2012) [...]"

- p. 2, l. 18-19: "Thus, both models reproduce equivalent "flushing" in both hemispheres, which is not confirmed by the available observations." What does it mean that the models are reproducing a phenomenon not confirmed by observation?

  *The data basis in the SH lowermost stratosphere is considerably less reliable than in the NH. Thus, they data may not be sufficient to constrain the models. Not only related to the amount of data in the respective SH seasons, but also in the case of the southern hemispheric autumn to substantially different observations from TOGA and the Whole Air sampler. The model results of respective hemispheric autumn lowermost stratosphere are quite similar with only a small vertical gradient in the $CH_2Br_2$ profiles, leading to a similar "flushing" on both hemispheres. We have slightly changed the sentence to show again that the comparison was made with limited observations in the LMS of the southern hemisphere*

  "[...] Thus, both models reproduce equivalent "flushing" in both hemispheres, which is not confirmed by the limited available observations. [...]"

- p. 6, l. 175: "with fixed emissions of the VSLS during the whole modeling period"
  What time resolution is the emission data, Annual or monthly climatology?

  *The emission inventory has a monthly seasonality, both when used in TOMCAT and CAM-Chem. In both subsections, we include this information:*

  In section 3.1:
  "[…] In this study, the VSLS emission scenario of Ordóñez et al. (2012), which includes monthly variability in emissions, was used with TOMCAT. […]"

  In section 3.2:
  "[…] As with the TOMCAT model, the monthly varying emission scenario of Ordóñez et al. (2012) was used, with fixed emissions of the VSLS during the whole modelling period (available from 2009–2019). […]"

- p. 7, l. 188-200:
  1) It would be better to separate this section as analysis methods.

     *We agree that a separation or naming of this part would be an improvement. Therefore, this part is now labeled as a subsection "4.1 Analysis methods" (thus, numbering of the following subsections changes as well).*

  2) From what data were the equivalent latitudes and temperatures calculated for the observational and model data analysis, respectively?

     *Equivalent latitudes and tropopause information (e.g., potential temperature at the local tropopause) along the flight tracks were derived with the Chemical Lagrangian Model of the Stratosphere (CLaMS) with underlying ECMWF reanalyzes (see section 3.3). We further expand the information in section 3.3:*

     "[…] In addition, local tropopause information along the flight tracks as well as equivalent latitude were derived using the Chemical Lagrangian Model of the Stratosphere (CLaMS) (e.g., Grooß et al., 2014) with underlying ECMWF reanalysis […]"

     *No information about the equivalent latitude was used for the models, only the latitude. Furthermore, the potential temperature difference to the tropopause was calculated from the potential temperature of the models relative to the climatological tropopause. Information about the climatological tropopause can be found in section 3.3 and about the potential temperature difference for the analysis in section 4.3 and 4.4.*

  3) Please clarify which sections latitude-altitude, $\theta$, $\Delta\theta$, and equivalent latitude coordinate systems are used for.

*In the newly introduced Section 4.1, we have added the sections in parentheses at the appropriate places to indicate where the corresponding coordinate systems are used.*

4) Please clarify how do you sample the model data, along the aircraft tracks or regional mean?

*Model data only from the years and months where observations are available were used (as written in the manuscript). The comparison was done using zonal mean model data. In addition, we tested regional mean model data, e.g., model data only in predefined radius around observations data. However, there was no noticeable difference to results using the zonal mean model data.*
*We extract the information on model results from Section 4.2 "Model results are used only for the years and months when observations are available" and include this information, as well as on the use of zonally averaged model observations, in the newly introduced Section 4.1.*

"[…] In all sub-analyses (Sect. 4.2 – 4.4), the observations are compared with the model data. The model results are only used for the years and months for which observations are available and have been zonally averaged (consistent with Keber et al., 2020). […]"

- p. 8, l. 221: "While the distribution of CH2Br2 in hemispheric spring is quite similar, …" What is similar to what, the distributions of NH and SH?

*Correct, the focus here is the comparison of the distribution of the LMS in the NH and SH. The term "… hemispheric spring…" also points to this. The second part of the sentence says "…, the distribution in hemispheric autumn differs with smaller values in the SH compared to the NH", also making clear that this a comparison of NH and SH. We have expanded the sentence so that it now reads as follows:*

"[…] While the distribution of CH$_2$Br$_2$ in hemispheric spring is quite similar in both hemispheres, […]"

- p. 8, l. 222: "… differs with smaller values in the SH compared to the NH" How much is the difference?

*A direct bin to bin comparison (in terms of exact values) is difficult because they are not in tropopause relative coordinates in this representation. Nonetheless, the difference in hemispheric autumn is up to roughly 0.3 ppt, whereas difference is spring is mostly well below 0.1 ppt. We include an additional sentence with the rough difference (0.3 ppt) at the subtropical and extratropical tropopause.*

"[…] differs with smaller values in the SH compared to the NH. Mixing ratios above the subtropical and extratropical tropopause are up to 0.3 ppt smaller in the SH. […]"

- p. 9, l. 258: "Ordonez et al. (2012) already identify some issues regarding…" Please clarify what issues were identified?

*The issues, which were already listed in the publication of Ordóñez et al. (2012), are mentioned in this and next sentences. Especially the very few observations in the Southern Hemisphere from 40 to 90° S (which are only from boral autumn) effecting the missing seasonality of the VSLS emission fluxes. Furthermore, they stated that a comprehensive parametrization of processes at the sea ice interface would also be required for a better representation of emission in polar regions. As tackling the issues with the emission inventory in detail would be beyond the scope of this publication, we would leave the short introduction to the issues to these two sentences but slightly re-writing the sentences as followed.*

"[…] Ordóñez et al. (2012) already identified the issue regarding the emission flux estimates in the SH as a consequence of missing aircraft observations in the SH (especially south of 40°S for all seasons) […]"

- p. 9, l. 266: "Nonetheless, the NH shows larger values in autumn and winter compared to spring and summer …" How much?

*Differences between winter and summer tropospheric values can be as high as 5 ppt (high latitudes and lowest altitude), but are usually around 1 ppt. The differences between autumn and spring are in the range of 0.5 to 1 ppt, also at high latitudes. Since the near ground tropospheric observations are not the focus of this work, we would prefer not to list the values here. Further, near ground tropospheric observation may not be representative as aircraft campaigns were partly near coastlines. This is addressed a little later in the paragraph. Nevertheless, we would supplement this sentence with the following:*

"[…] Nonetheless, the NH shows larger values in autumn and winter compared to spring and summer, a feature that is captured by both models, although the modeled wintertime maxima are more pronounced than in the observations and much less pronounced in autumn.
Near-ground observations, however, may not be representative as they are largely from coastal areas. […]"

- p. 10, l. 308-309 "Both models quantitatively reproduce the larger CH2Br2 values in hemispheric winter and spring and smaller values in summer and autumn (see Fig. S3)." Which latitude bands does this statement refer? The models do not look like reproducing the observed seasonality of CH2Br2 in low latitudes.

*You are correct. Larger values in hemispheric spring are reproduced in high and mid latitudes. So, this statement is true for these two latitude bands, which should be stated in the text. We include this in this sentence by writing:*

"[…] Both models qualitatively reproduce the larger $CH_2Br_2$ values in hemispheric winter and spring and smaller values in summer and autumn at high and mid latitudes (see Fig. S3) […]"

- p. 12, l. 382: "hemispheric autumn profiles of CH2Br2 and CHBr3 are less similar than in hemispheric spring." What is less similar, profiles of CH2Br2 and CHBr3 or profiles in the NH and SH?

  *What is meant are the profiles of NH and SH. We rewrite this sentence as follows to make this clearer:*

  "[…] Figure 7 shows hemispheric autumn profiles of $CH_2Br_2$ and $CHBr_3$ with less similarity of SH and NH profiles than in hemispheric spring for both compounds. […]"

- p. 13, l. 386; "Differences between the hemispheres become larger on lowest levels above the dynamical tropopause, i.e., in the ExTL." Compared to what, do the inter-hemispheric differences become larger?

  *In this section, we compared SH and NH profiles of $CH_2Br_2$ in hemispheric autumn. We worked our way from tropospheric values through the tropopause to the lowermost stratosphere of the respective hemisphere. As we discussed the hemispheric differences at the tropopause right before this sentence, the larger differences in the ExTL were compared to them and thus the interhemispheric difference become larger. We slightly changed the sentence to.*

  "[…] Differences between NH and SH autumn become larger on lowest levels above the dynamical tropopause, i.e., in the ExTL. […]"

- Figures 2 and 3: What is the reason for missing values in the lower stratosphere in the TOMCAT model? (Second row).

  *Missing data is the result of the vertical resolution of the TOMCAT model combined with the intervals chosen to bin the data.*
  *Bin size decreases logarithmically with increasing altitude and thus lower pressure. Binning at this altitude is between about 80 and 68 hPa, thus a bin size of about 12 hPa. No model data appear to be available for this interval, but this does not affect the comparison with the observed data as observations are only above 100 hPa. We included a short notification in the figure caption.*

  "[…] The slightly coarser vertical resolution of TOMCAT combined with the bin size leads to missing TOMCAT data between 68 and 80 hPa. […]"

- Figures 5 and S1: description of line colors are missing.

  *We included missing color description in the caption of Figure 5 and S1 as followed:*

[revised manuscript text omitted]